# Sustainability Assessment of Annual and Permanent Crops: The Inspia Model

**Paula Trivino-Tarradas** [1,2,*], **Manuel R. Gomez-Ariza** [3], **Gottlieb Basch** [2,4] and **Emilio J. Gonzalez-Sanchez** [1,2,3] 

1   ETSIAM, Campus de Rabanales, Universidad de Córdoba, Ctra, Nacional IV, km. 396, 14014 Córdoba, Spain; emilio.gonzalez@uco.es
2   European Conservation Agriculture Federation (ECAF). Rue de la Loi 6 Box 5, 1050 Brussels, Belgium
3   Asociación Española de Agricultura de Conservación Suelos Vivos (AEAC.SV), Centro IFAPA Alameda del Obispo, Av. Menéndez Pidal s/n, 14004 Córdoba, Spain; mgomez@agriculturadeconservacion.org
4   Institute of Mediterranean Agricultural and Environmental Sciences (ICAAM), Universidade de Évora, Polo da Mitra, Ap. 94, 7006-554 Évora, Portugal; gb@uevora.pt
*   Correspondence: ptrivino@uco.es; Tel.: +34-957-21-84-56

**Abstract:** The Initiative for Sustainable Productive Agriculture (INSPIA) project promotes best management practices for agriculture, to enhance the provision of ecosystem services through better stewardship of soil and water resources while ensuring high levels of productivity. This paper presents the INSPIA methodology for the assessment of sustainability and for guiding farmers on strategic decision-making at farm level, applicable to any kind of cropland. The methodology is based on the application of 15 best management practices, which are assessed through a set of 31 basic sustainability indicators that cover the economic, social and environmental dimensions both agreed by a panel of experts. Basic indicators are then grouped into 12 aggregated indicators, to build the final INSPIA composite index. The INSPIA methodology provides farmers and advisers with a tool to understand sustainability and which, to a certain extent, serves to improve performance toward sustainability. Results are presented in three different ways: a bar diagram with the whole set of basic indicator-values; a pie chart representing the sustainability split in the aggregated indicators; and a final sustainability index. In the medium and long term, the INSPIA methodology can help to monitor and assess agricultural and environmental policy implementation, as well as help improve its decision-making processes in the future.

**Keywords:** sustainable agriculture; best management practices; sustainability indicators; composite index; strategic decision-making

## 1. Introduction

Agriculture faces many challenges. Not only does it have to produce more food, feed and other raw materials to satisfy the increasing demands of the growing population, it also must contribute to economic prosperity and social well-being, while protecting natural resources. Farming is particularly required to demonstrate its efficiency, as agriculture is a user of finite resources [1]. In this regard, there is a growing interest in European society, concerned with improving the relationship between agricultural production and the environment [2]. Indeed, agricultural sustainability is a cross-cutting EU priority, through ecosystems conservation, biodiversity improvement, preservation of water and soil quality in agriculture-related ecosystems [3–5].

There is a broad consensus that agricultural sustainability implies meeting the needs of the present, without compromising on the needs of the future generations with regard to food, feed and

fibre production [6]. Since 1987, when the first discussions on sustainable agriculture emerged [7], numerous attempts to measure agricultural sustainability, ranging from the farm-level to regional or national-level applications, are found in the scientific literature [8]. In this context, innovative integrated approaches such as the life cycle assessment (LCA) tools seem to make a great contribution to sustainability evaluations [9]. These tools help in the transition towards more sustainable production and consumption patterns [10] and can provide additional information about environmentally sustainability in a wider and long-term perspective [11]. Due to the inherent multi-dimensional nature of the sustainable development concept, it is accepted that agricultural sustainability should involve three pillars: economic (economically viable for the survival of the farms); social (keep or improve farmers' life and working conditions); and environmental (protect and even enhance the environment and protecting the natural capital), [12–20].

There are some successful initiatives to monitor specific issues in agricultural systems, such as soil quality [21]. However, there is a lack of consensus among stakeholders on the criteria to take into account for agricultural sustainability assessment [16]. To the best of our knowledge, until the last two decades, there have not been many approaches to agriculture sustainability assessment based on the social, economic and environmental dimensions [22–24]. In most cases, attention was just focused on one of the three aspects (economic, social, environmental) [20,24–26]. For improving conventional farming performance, holistic approaches are needed, and therefore the Initiative for Sustainable Productive Agriculture (INSPIA) promotes a set of comprehensive best management practices (BMPs).

Farmers are valuable to society [27], since they are considered to be the largest natural-resource managers in worldwide ecosystems. Certainly, a major part of biodiversity in agriculture depends highly on how agricultural land is managed [3]. Therefore, BMPs in agriculture play a major role not only in biodiversity conservation, but also in other natural capital aspects. BMPs deliver ecosystem services while helping farmers deliver food production [5]. Therefore, transfer of BMP technology to farmers is essential so they can enhance the environment whilst producing quality food and fibre [28,29].

Recent European agriculture policies procure economic profitability, environmental safety and social fairness. In fact, the Common Agricultural Policy (CAP) has experimented with a "greening" process, from the first agri-environmental measures in the 1992 reform, to the current practices that benefit the environment and the climate within the period 2014–2020. The current greening obligations in the CAP's Pillar I (2014–2020), and the disposition towards an even greener CAP, are shown in the different forums where the future of the CAP is discussed and provided in the Communication COM (2017) 713 "The Future of Food and Farming" [5] and in the proposal for the new regulation (COM(2018) 392 final) [30]. It is very likely that there will be an increased demand to monitor and measure agricultural sustainability through flexible holistic initiatives [31–33], that need to consider the implementation of result-oriented schemes [34,35].

The INSPIA project [36] aims to provide a road map to sustainable agriculture through the implementation of 15 BMPs and the measurement and monitoring of progress with a set of 31 defined indicators. In the framework of the INSPIA project, the aim of this paper is to propose a methodology for assessing sustainability at farm level by providing a final composite index. This methodology has been tested in 59 private farms distributed throughout Europe that belong to the INSPIA network.

Future research on sustainability assessment should be focused on merging INSPIA assessment methodology with the LCA tools. Adopting both will provide stakeholders outcomes more adherent to the realities under study, compensating some lacking aspects, derived from the use of multi-criteria decision analysis [9,11].

## 2. Materials and Methods

### 2.1. The Method for Selecting the BMPs and Indicators

The BMPs and indicators have been selected by a panel of experts formed by a multidisciplinary team. The authors of this article have been regularly involved in international research, development

and innovation (RDI) projects for over 20 years. The network created over that long period facilitated the selection of the international panel of experts. Not only people of different disciplines from academia (agronomists, weed sciences, economists, sociologists, environmentalists), but also farmers, their representatives and non-governmental organisations (NGOs) were represented in the panel: 5 representatives of farmers; 4 members from European universities; 4 members from public research stations; 4 representatives of non-profit making associations; 3 from the private sector. The members of this panel were chosen on the basis of their experience and knowledge on each agricultural sustainability dimension. INSPIA methodology is in agreement with Bockstaller and Girardin [37], who recognised that expert judgment as one of the validation procedures to meet the quality criteria in the selection of indicators.

For the selection of the BMPs and indicators, a thorough literature review of existing methodologies was made. In total, 6 meetings were organised for the setting of the indicators, the BMPs and the procedure to build the INSPIA composite index (Figure 1). As a result of these meetings, the basic indicators, their aggregation, their normalisation methods and the weight assignments were selected. The INSPIA BMPs were agreed unanimously by the 20 members of the panel.

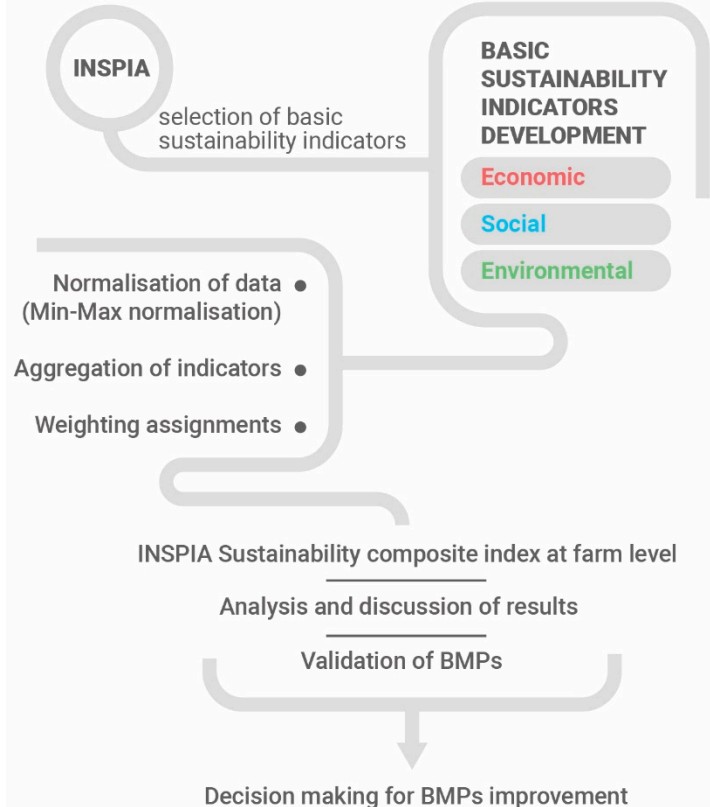

**Figure 1.** Structure and functioning of the Initiative for Sustainable Productive Agriculture (INSPIA) indicator-based sustainability assessment process.

### 2.2. INSPIA—Best Management Practices

The sustainable performance of a farm is affected by the number of BMPs implemented. The INSPIA index result and its basic indicators' value, are also related to the number of BMPs followed by a farmer, as will be shown in the results.

### 2.3. INSPIA Sustainability Indicators

Indicators provide the guidelines for farmers steering farm management towards an improvement of sustainability [38–41]. The more indicators are considered, the more accurate is the sustainability index. According to the literature, an acceptable procedure is to build a sustainability index, which

should be based on the following steps; selection of indicators and their development, as well as normalisation, aggregation and weighting [19,20,25,31,33,39,41–44].

### 2.3.1. Indicator Development

Basic indicators have been developed to meet the scientific standards, through a sound method of collecting data in the fields [15,31,45,46]. Some of the indicators selected for INSPIA are based on previous initiatives that evaluate and analyze agricultural sustainability, for example, indicIADes from the Institute de l'Agriculture Durable [47], INDIGO® [48], Sustainability Assessment of Farming and the Environment-SAFE [45], Multicriteria Assessment of the Sustainability of cropping Systems-MASC [22], DEXi Pest Management-DEXiPM [31], Sustainability Assessment in Food and In Agricultural Systems-SAFA [41], SOSTARE [33], Sustainable Agri-Food Evaluation Methodology-SAEMETH [49].

The selection of basic indicators was carried out on the basis of reliability criteria and applicability, permitting their operational calculation since the information can be obtained directly from farmers. Farmers' endorsement is considered essential to ensure the acceptability of the indicator selection.

The selection of INSPIA indicators fulfils the three types of validation: (i) 'design validation', since the indicators are scientifically referenced and possess a degree of accuracy [46,50]; (ii) 'output validation', because there is soundness of the indicators output, and usefulness for potential users [50]; and finally (iii) 'end-use validation' since they are useful, and are used by the decision aid tool [37,51].

### 2.3.2. Normalisation of Indicators

As illustrated by Singh et al., [24], a given indicator does not provide relevant information unless there is a reference value associated with the indicator itself. Since basic indicators have been calculated using different measurement units, transforming these basic indicators into a non-dimensional value is paramount to make them operational. In INSPIA's case, among the different normalisation techniques described in the literature, the panel of experts decided to employ the "min-max" normalisation method [52]. Therefore, the basic indicators' values would range between 0 and 1, and be scored by using valuation functions.

The conditions of the location of the farms need to be taken into account when normalising indicators [53]. In INSPIA, some indicators have been adapted to the participating countries, using different ranges (maximum and minimum value) depending on the country itself, which will correspond to 0 and 1 on the normalisation. The choice of quantitative thresholds used for some basic indicators is crucial and will partly determine the value of the assessment results. For INSPIA, the minimum and the maximum thresholds, to normalise the indicator ranges in the different countries, stem from the literature, knowledge and insight of experts.

### 2.3.3. Weighting of Indicators

The weighting of indicators are needed for the subsequent aggregation operations. Among the existing methods for allocating weight described in the literature [19,54] there are: (i) equal weighting; (ii) statistic-based weighting; and (iii) public/expert opinion-based weighting, "professional opinion".

As stated previously, the panel of experts established the selected methodology for INSPIA. Therefore, although assigning a weight to indicators depends on subjective scoring [24,39,55], a transparent and participatory method was followed within the INSPIA panel of experts, where the guidelines for assigning weights to indicators were made under the assumption that the three sustainability dimensions are equally relevant.

Members of the panel were given a total of 100 points to be distributed among indicators in each level class (aggregated and basic ones) according to their relevance. The higher the indicator importance, the more points were allocated to it. The final score comes from the rounding up to ten of the arithmetic mean. Weighting indicators, facilitates the decomposition of a certain problem into a

hierarchical structure and ensures that both qualitative and quantitative aspects of indicators count in the assessment process.

### 2.3.4. Aggregation of Indicators

Once the indicators are weighted and transformed into component scores (aggregated indicators), these scores are able to be aggregated into a composite score at level 1 (aggregation of basic sustainability indicators), and so on for level 2. Aggregated indicators for level 2 result from the combination of aggregated indicators at level 1.

The panel of experts agreed upon the process and defined the hierarchical multi-criteria structure for the basic indicators. The procedure of indicator aggregation determines the type of compensation, also called "marginal substitution rate" in the economic literature, among indicators [56]. As well as for the previous normalisation and weighting operations, various methods exist for aggregation [19,54]. By far, the most widespread linear aggregation is the summation of normalised indicators which corresponds to the "additive aggregation methods" [19], which assume the total compensation among the indicators involved [57]. This was the method chosen by the INSPIA panel.

### 2.4. INSPIA Sustainability Composite Index

As stated by Gómez-Limón and Riesgo [58] the difficulty of interpreting the multi-dimensional set of indicators and aggregated indicators can be overcome by aggregating them into a single index or composite index. In addition, having a final index composed of more than one indicator eases the understanding of complex information by non-experts [59], and would likely impact the system monitored sustainability [24,60]. In INSPIA, all sustainability dimensions are equally important to agricultural sustainable development, and cannot be substituted by each other, as agreed upon by many authors [39,61].

### 2.5. INSPIA Online Tool

The INSPIA online calculator is available at www.inspia-europe.eu. Nowadays, its geographical scope covers 59 cropped land farms located in Belgium, Denmark, France, Germany and Spain (Figure 2); both annual and permanent croplands are represented (winter cereals, oilseeds, legume and root crops for annual crops and olive trees and vineyards for permanent crops), and its thematic scope is economic, social and environmental.

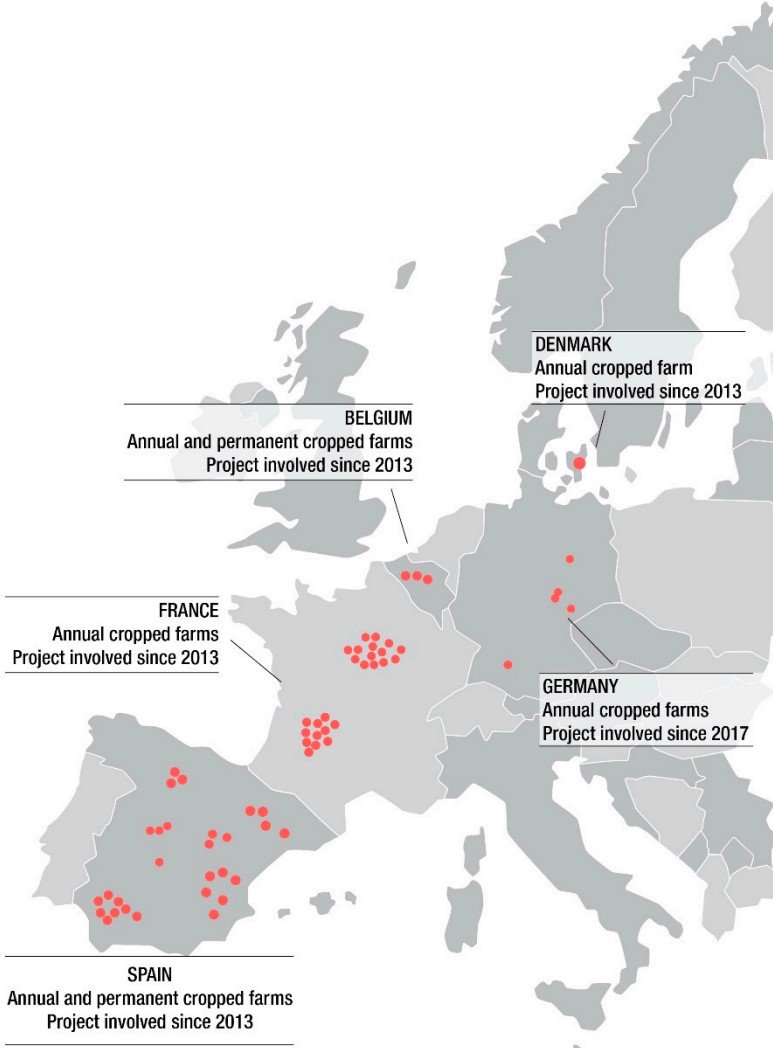

**Figure 2.** INSPIA farm network.

## 3. Results

The INSPIA panel of experts proposed the following results concerning BMPs, basic sustainability indicators, their aggregation and weighting, and the final INSPIA composite index.

### 3.1. Set of INSPIA BMPs

INSPIA's BMPs list is presented in Table 1. The BMPs are comprehensive and deal with the essential components of sustainability. Some of the BMPs are related to the soil and crop management, while others deal with agricultural input-management. Other BMPs are aimed at environmental improvement and natural capital protection. Social aspects and farmers' welfare are also included in the BMPs.

The implementation of INSPIA's BMPs would result in better soil, water and air quality, whilst keeping or even improving yields. Also, BMPs would help farmers optimise the use of inputs, which can result in a more profitable agricultural system [62].

**Table 1.** List of INSPIA's best management practices.

| BMP | Name of the INSPIA BMP |
|---|---|
| BMP 1 | Use permanent soil cover (green cover or residue cover). |
| BMP 2 | Use of minimum soil disturbance practices. |
| BMP 3 | Use of groundcovers (in permanent crops). |
| BMP 4 | Perform suitable crop rotation/diversification. |
| BMP 5 | Perform farming operations following the contour lines. |
| BMP 6 | Fertilize according to soil deficiencies and crop needs. |
| BMP 7 | Plant protection products use according to Integrated Pest Management (IPM) principles. |
| BMP 8 | Use of modern technologies for applications (precision agriculture). |
| BMP 9 | Optimise irrigation timing and rate (considering soil water content, water holding capacity in the soil, and crop requirements in relation to evapo-transpiration). |
| BMP 10 | Optimised use of pesticides (correct dose and appropriate product) |
| BMP 11 | Implementation of field margins and buffer strips with diversity of plant species. |
| BMP 12 | Establish and maintain riparian buffers. |
| BMP 13 | Build retention structures across slopes to reduce length of plots (fascines, vegetative buffers). |
| BMP 14 | Point source prevention of PPP (pesticide) pollution on the farm (establish areas to fill and clean sprayers and manage containers). |
| BMP 15 | Perform optimised waste management (packaging, crop residues, effluents, pesticide containers, etc.). |

## 3.2. INSPIA Sustainability Indicators

The performance of the BMPs is monitored through a tailored set of 31 open-source basic indicators (Table 2), that cover the three main thematic dimensions of sustainable development [63]. The correspondent calculation formula and definition of each basic indicator can be found in the website www.inspia-europe.eu.

**Table 2.** Set of INSPIA basic sustainability indicators.

| No. | INSPIA Basic Sustainability Indicators | Units | Sustainability Dimension (Thematic Scope) |
|---|---|---|---|
| 1 | Net income per ha | €/ha | |
| 2 | Net income per annual work unit (AWU) | €/AWU | |
| 3 | Production cost per ha | €/ha | |
| 4 | Yield | | |
| 5 | N Productivity | kg/kg | |
| 6 | P Productivity | kg/kg | Economic dimension |
| 7 | Irrigation water application | $m^3$/ha | |
| 8 | Water productivity | kg/$m^3$ | |
| 9 | Energy balance | MJ/ha | |
| 10 | Energy efficiency | MJ/MJ | |
| 11 | Energy productivity | kg/MJ | |
| 12 | Working hours per ha | h/ha | |
| 13 | Satisfaction index | - | |
| 14 | Farmers' training levels | - | Social dimension |
| 15 | Risk of abandonment of agricultural activity | - | |
| 16 | Soil tillage index | - | |
| 17 | Soil cover rate | - | |
| 18 | Organic matter | - | |
| 19 | Soil erosion risk | % | |
| 20 | Crop diversity | - | |
| 21 | Crop rotations | - | |
| 22 | N Balance | kg N/ha | |
| 23 | N Efficiency | kg/kg | |
| 24 | P Balance | kg P/ha | |
| 25 | P Efficiency | kg/kg | Environmental dimension |
| 26 | GHGs Balance | $CO_2$eq/ha | |
| 27 | GHGs per kg | Kg $CO_2$eq/kg | |
| 28 | Natural area | % | |
| 29 | Biodiversity structures | - | |
| 30 | Buffers and security areas | % | |
| 31 | PPP management | - | |

The performance of the INSPIA BMPs is determined annually in each farm, and the evolution of the indicators shows the effectiveness of the BMPs' implementation (Table 3). The results of the indicators are intended to identify which practices need to be improved in the next seasons in order to enhance farm performance (Figure 3).

**Table 3.** Matrix connecting INSPIA best management practices (BMPs) with basic indicators.

| BMP | 1 | 2 | 3 | 4 | 5 | 6 | 7 | 8 | 9 | 10 | 11 | 12 | 13 | 14 | 15 | 16 | 17 | 18 | 19 | 20 | 21 | 22 | 23 | 24 | 25 | 26 | 27 | 28 | 29 | 30 | 31 |
|---|---|---|---|---|---|---|---|---|---|---|---|---|---|---|---|---|---|---|---|---|---|---|---|---|---|---|---|---|---|---|---|
| 1 | X | X | X | X | X |   | X | X | X | X | X | X | X | X | X | X | X | X | X | X | X | X | X |   |   | X | X |   | X |   |   |
| 2 | X | X | X | X | X |   | X | X | X | X | X | X | X | X | X | X | X | X | X |   |   | X | X |   |   | X | X |   | X |   |   |
| 3 | X | X | X | X | X |   | X | X | X | X | X | X | X | X | X | X | X | X | X | X |   | X | X |   |   | X | X |   | X | X |   |
| 4 | X | X | X | X | X | X | X | X | X | X | X | X | X | X | X | X | X | X | X | X | X | X | X | X | X | X | X |   | X |   |   |
| 5 |   |   |   |   |   |   |   |   |   |   |   |   |   | X |   |   |   | X |   |   |   |   |   |   |   |   |   |   |   |   |   |
| 6 | X | X | X | X | X | X |   | X | X | X | X | X | X | X |   |   |   | X |   |   |   | X | X | X | X | X | X |   |   |   |   |
| 7 | X | X | X | X |   |   |   | X | X | X | X | X | X | X |   |   |   |   |   |   |   |   |   |   |   | X | X |   |   |   | X |
| 8 | X | X | X | X | X | X | X | X | X | X | X | X | X | X |   |   |   |   |   |   |   | X | X | X | X | X | X |   |   |   | X |
| 9 | X | X | X | X |   |   | X | X | X | X | X | X | X | X |   |   |   | X | X |   |   |   |   |   |   | X | X |   |   |   |   |
| 10 | X | X | X | X |   |   |   | X | X | X |   |   | X | X |   |   |   |   |   |   |   |   |   |   |   | X | X |   |   |   | X |
| 11 | X | X | X | X |   |   |   | X | X | X | X | X | X | X |   | X | X | X | X |   |   |   |   |   |   | X | X | X | X | X |   |
| 12 | X | X | X | X |   |   |   | X | X | X | X | X | X | X |   | X | X | X | X |   |   |   |   |   |   | X | X | X | X | X |   |
| 13 |   |   |   |   |   |   |   |   |   |   |   |   |   | X |   |   |   | X |   |   |   |   |   |   |   |   |   | X | X | X |   |
| 14 |   |   |   |   |   |   |   |   |   |   |   |   | X | X |   |   |   |   |   |   |   |   |   |   |   |   |   |   |   |   | X |
| 15 |   |   |   |   |   |   |   |   |   |   |   |   | X | X |   |   |   |   |   |   |   |   |   |   |   |   |   |   |   |   | X |

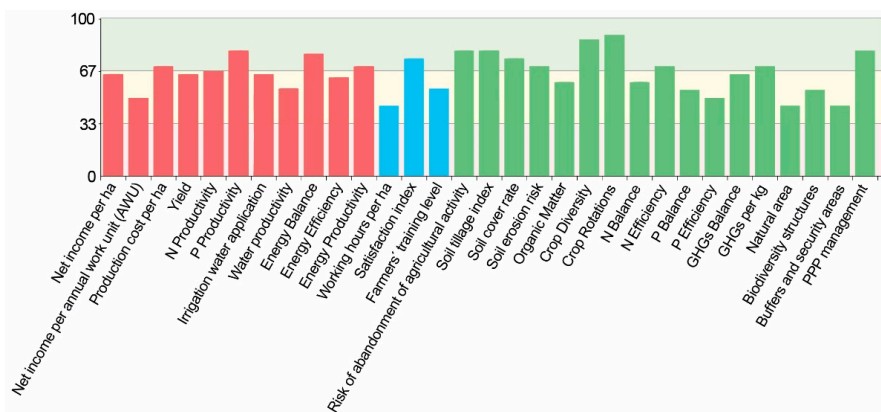

**Figure 3.** INSPIA graphical and numerical outcomes example on sustainability basic indicators.

### 3.3. INSPIA Aggregated Indicators

In the aim of reaching a final composite index, combining the basic indicators into aggregated components was required, which ended up generating the three sustainability dimensions that compose the INSPIA sustainability index: economic, social and environmental (Figure 4). This process involves choosing the functional operational form, in which indicators are built at levels 1 and 2, as can be seen in Figure 5.

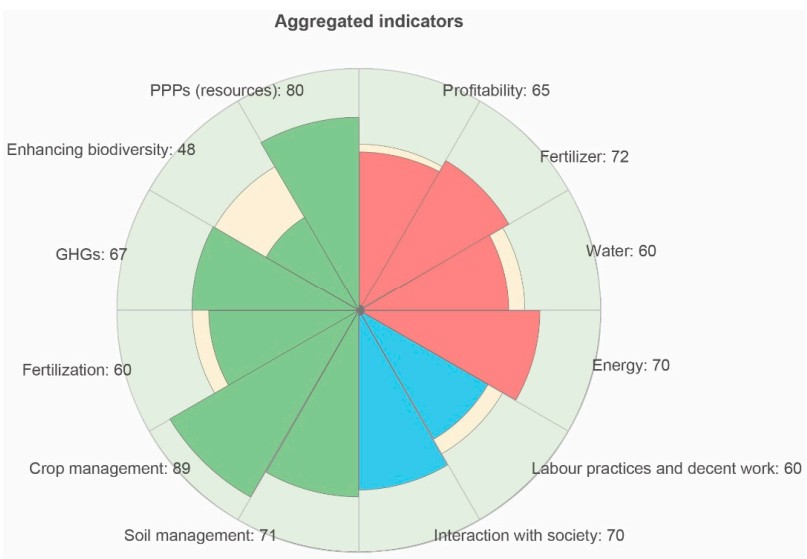

**Figure 4.** INSPIA graphical and numerical outcomes example on sustainability aggregated indicators.

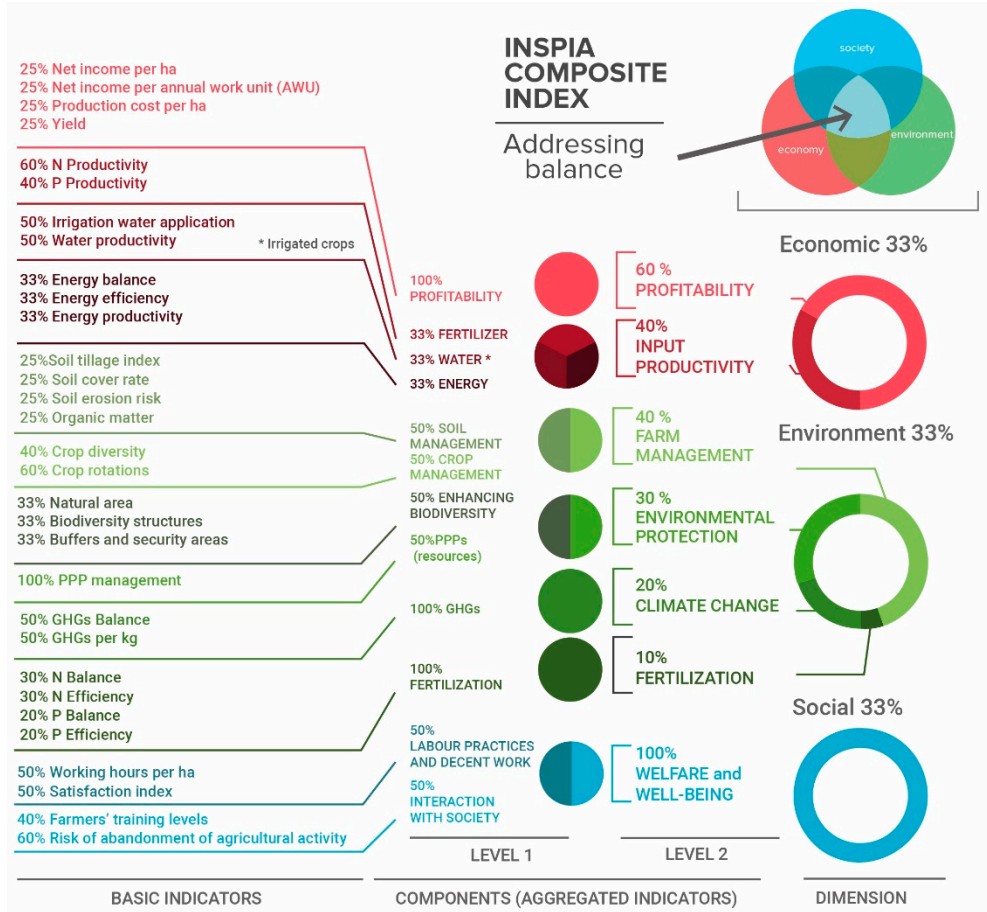

**Figure 5.** Weighting sustainability indicators, aggregated indicators (level 1 and 2), and sustainability dimensions.

*3.4. INSPIA Sustainability Index*

INSPIA's composite index is the result of the arithmetic mean of the three aggregated indicators, corresponding to the economic, social and environmental dimensions. The three dimensions are given the same weight, and lead on to an even distribution for providing a well-balanced sustainability index. Therefore, INSPIA index can be calculated as follows:

$$INSPIA\ Sustainability\ Index = \frac{\sum Sustainability\ dimensions}{3} \tag{1}$$

Sustainability dimensions = Economic, social and environmental.

Values range from the lowest to the highest, depending on what farmers perform in the field and, in the end, depending on the level of implementation of the INSPIA BMPs at the farm. For instance, an optimal set of indicator values is a set of uniformly high values. A high average score, but one that includes very low values on some basic indicators, is sub-optimal and not sustainable, even though these are steps in the right direction. Therefore, basic sustainability indicators should improve with the BMPs' implementation on the farm. For INSPIA, the score 0 stands for the worst case, whereas the score 100 stands for the best state. The score 67 implies a threshold for sustainable agricultural practices [64]. This means that all results, from 67 and above it, can be regarded as sustainable (Figure 6).

**INSPIA Composite Index**

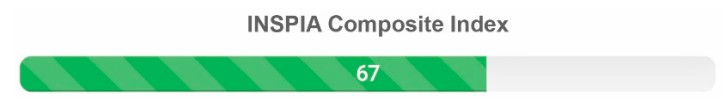

**Figure 6.** INSPIA graphical and numerical outcomes example on composite index.

## 4. Discussion

### 4.1. Set of BMPs Selection

Most of the INSPIA BMPs promote biodiversity and contribute to safeguarding the soil and water resources on which sustainable agricultural productivity depends, whilst delivering ecosystem services. This implies a holistic and sustainable agriculture system approach, based on the combination of:

- Conservation Agriculture (CA), fundamentally driven by BMP1, BMP2, BMP3 and BMP4. The principles of CA are: (i) minimum soil disturbance; (ii) permanent soil cover; and (iii) crop rotations [62,65–67]. The implementation of the CA principles plays a major role in the mitigation and adaptation towards climate change, since it implies a reduction of greenhouse gas (GHGs) emissions by fixing $CO_2$ from the atmosphere as soil organic carbon [68]. Moreover, CA systems deliver ecosystem services, as a result of improved conditions in the soil volume used by plant roots, and by enhanced functional agrobiodiversity [65,69].
- Integrated Pest Management techniques, addressed in BMP7 and BMP10, which are aligned with the framework for Community action to achieve the sustainable use of pesticides established in the Directive 2009/128/EC of the European Parliament and of the Council of 21 October 2009 [70].
- The use of site-specific crop management, such as precision agriculture, is associated with BMP8.
- Input optimisation, conveyed in BMP6, BMP9 and BMP10.
- Habitat enhancement and environmental protection, addressed in BMP5, and from BMP11 to BMP15.
- And more biodiversity, which is related to a higher carbon sequestration and erosion control [71]. Indeed, Overmars et al., [3] concluded that less biodiversity was associated with intensive tillage of the soil. Biodiversity is promoted in many BMPs such as BMP1, BMP2, BMP3, BMP4, BMP7, BMP8 and, finally BMP10, BMP11, BMP12 and BMP13.

INSPIA's BMPs are aligned with the United Nations' Sustainable Development Goals (SDGs), which were launched in the 2030 Agenda for Sustainable Development [63]. The SDGs also integrate the three dimensions of sustainable development. INSPIA contributes to 9 out of the 17 SDGs; SDG2-'Zero hunger'; SDG4 'Quality education'; SDG6 'Clean water and sanitation'; SDG7 'Clean energy'; SDG8 'Good jobs and economic growth'; SDG9 'Innovation and infrastructure'; SDG12 'Responsible consumption'; SDG13 'Protect the planet' and SDG15 'Life on land'. Table 4 shows how INSPIA BMPs contribute to the to the SDGs. As an example, in relation to SDG13: related to climate action, INSPIA encourages the delivery of certain soil management practices that make a measurable contribution to reduce carbon emissions, and contribute to a more climate-resilient agriculture. Certainly, INSPIA is responding to the global concern of mitigating and adapting to climate change. Conservation Agriculture is highlighted in the international initiative "4 per 1000", launched by the French Ministry of Agriculture at the COP21. This ambitious initiative will guide stakeholders not only towards environmental improvements but also to creating greener jobs and incomes hence ensuring sustainable development [72].

**Table 4.** INSPIA BMPs' contribution to the United Nations' Sustainable Development Goals (SDGs).

| | SDGs | | | | | | | | |
|---|---|---|---|---|---|---|---|---|---|
| | 2 | 4 | 6 | 7 | 8 | 9 | 12 | 13 | 15 |
| BMP | Zero Hunger | Quality Education | Clean Water and Sanitation | Affordable and Clean Energy | Decent Work and Economic Growth | Industry, Innovation and Infrastructure | Responsible Consumption and Production | Climate Action | Life on Land |
| 1 | X | X | X | | | | X | X | X |
| 2 | X | X | X | X | X | | X | X | X |
| 3 | X | X | X | | | | X | X | X |
| 4 | X | X | | X | X | | X | X | X |
| 5 | | X | X | | | | | | X |
| 6 | X | X | X | X | X | | X | X | X |
| 7 | X | X | X | X | X | | X | X | |
| 8 | X | X | X | X | X | X | X | X | |
| 9 | X | X | X | X | X | X | X | X | |
| 10 | X | X | X | X | X | X | X | X | X |
| 11 | | X | X | | | | | | X |
| 12 | | X | X | | | | | | X |
| 13 | | X | X | | | | | | X |
| 14 | | X | X | | | | X | | X |
| 15 | | X | X | | | | X | | |

## 4.2. Selection of Indicators (Basic, Aggregated and Index)

As was stated in the methodology section, the INSPIA method is based on 31 basic indicators. Table 5 shows the methodologies that share some indicators with INSPIA, while Table 6 shows the level of agreement between different methodologies with INSPIA.

**Table 5.** List of sustainability indicators used by INSPIA and other sustainability assessment methodologies. Colours correspond to the INSPIA dimensions: red: economic; green: environment; blue: social.

| | INSPIA Basic Sustainability Indicators | Other Sustainability Methodologies |
|---|---|---|
| 1 | Net income per ha | SAFE/DEXiPM/MOTIFS[1]/Gómez-Limón and Sanchez-Fernandez, 2010/IDEA[2]/MASC/MASC 2.0.[3]/RISE 3.0.[4]/SAEMETH/OECD/SOSTARE simplified |
| 2 | Net income per annual work unit (AWU) | MOTIFS/IDEA/MASC/MASC 2.0./OECD/SOSTARE simplified |
| 3 | Production cost per ha | DEXiPM/IDEA/MASC/MASC 2.0./OECD/SOSTARE simplified |
| 4 | Yield | MOTIFS/IDEA/MASC/MASC 2.0./RISE 3.0./SOSTARE simplified |
| 5 | N Productivity | DEXiPM/RISE 3.0./SAEMETH/OECD/SOSTARE simplified/IDEA/SAFE |
| 6 | P Productivity | DEXiPM/IDEA/MASC/MASC 2.0./RISE 3.0./SAEMETH/OECD/SOSTARE simplified/SAFE |
| 7 | Irrigation water application | SAFE/DEXiPM/Gómez-Limón and Sanchez-Fernandez, 2010/MOTIFS/MASC/MASC 2.0./RISE 3.0./SAFA/SAEMETH/OECD |
| 8 | Water productivity | Gómez-Limón and Sanchez-Fernandez, 2010/MASC/MASC 2.0./RISE 3.0./OECD/SOSTARE simplified/SAFE |
| 9 | Energy balance | SAFE/DEXiPM/Gómez-Limón and Sanchez-Fernandez, 2010/IDEA/MASC/MASC 2.0./RISE 3.0./SOSTARE simplified/INDIGO®/SAFE SAEMETH |
| 10 | Energy efficiency | DEXiPM/MOTIFS/IDEA/MASC/MASC 2.0./RISE 3.0./SOSTARE simplified/INDIGO®/SAFE/SAEMETH |
| 11 | Energy productivity | DEXiPM/MASC/MASC 2.0./RISE 3.0./SOSTARE simplified/INDIGO®/SAFE/SAEMETH |
| 12 | Working hours per ha | DEXiPM/Gómez-Limón and Sanchez-Fernandez, 2010/MASC 2.0./RISE 3.0./SAFA |
| 13 | Satisfaction index | IDEA/SAFA |
| 14 | Farmers' training levels | SAFE/IDEA |
| 15 | Risk of abandonment of agricultural activity | SAFE |
| 16 | Soil tillage index | DEXiPM/RISE 3.0./SAFA/OECD/SOSTARE simplified/SAFE |
| 17 | Soil cover rate | SAFE/RISE 3.0./Gómez-Limón and Sanchez-Fernandez, 2010/OECD/SOSTARE simplified/INDIGO® |

**Table 5.** *Cont.*

| | INSPIA Basic Sustainability Indicators | Other Sustainability Methodologies |
|---|---|---|
| 17 | Soil cover rate | SAFE/RISE 3.0./Gómez-Limón and Sanchez-Fernandez, 2010/OECD/SOSTARE simplified/INDIGO® |
| 18 | Soil erosion risk | MASC/MASC 2.0./IDEA/Gómez-Limón and Sanchez-Fernandez, 2010/SAFA/SAFE/RISE 3.0./OECD |
| 19 | Organic matter | DEXiPM/MOTIFS/IDEA/MASC/MASC 2.0./RISE 3.0./SAFA/SOSTARE simplified/INDIGO® |
| 20 | Crop diversity | DEXiPM/MOTIFS/IDEA/MASC/MASC 2.0./RISE 3.0./SAFA/INDIGO® |
| 21 | Crop rotations | Gómez-Limón and Sanchez-Fernandez, 2010/RISE 3.0./SAFA/MASC/MASC 2.0./SAEMETH/SOSTARE simplified |
| 22 | N Balance | SAFE/DEXiPM/Gómez-Limón and Sanchez-Fernandez, 2010/RISE 3.0./SAFA/SAEMETH/OECD/SOSTARE simplified/INDIGO®/IDEA |
| 23 | N Efficiency | DEXiPM/MOTIFS/RISE 3.0./SAFA/SAEMETH/OECD/SOSTARE simplified/INDIGO®/IDEA |
| 24 | P Balance | SAFE/DEXiPM/Gómez-Limón and Sanchez-Fernandez, 2010/IDEA/MASC/MASC 2.0./RISE 3.0./SAFA/SOSTARE simplified/SAEMETH/OECD/INDIGO® |
| 25 | P Efficiency | DEXiPM/MOTIFS/IDEA/MASC/MASC 2.0./RISE 3.0./SAFA/SAEMETH/OECD/SOSTARE simplified/INDIGO® |
| 26 | GHGs Balance | DEXiPM/MASC/MASC 2.0./RISE 3.0./SAFA/OECD/SAEMETH |
| 27 | GHGs per kg | DEXiPM/MASC/MASC 2.0./RISE 3.0./SAFA/OECD/SAEMETH |
| 28 | Natural area | MOTIFS/RISE 3.0./SAFA/SOSTARE simplified/IDEA/SAEMETH |
| 29 | Biodiversity structures | MOTIFS/RISE 3.0./SAFA/SAFE |
| 30 | Buffers and security areas | MOTIFS/RISE 3.0./SAFA/IDEA/SAFE |
| 31 | PPP management | MOTIFS/Gómez-Limón and Sanchez-Fernandez, 2010/IDEA/MASC/MASC 2.0./RISE 3.0./SAFA/SAEMETH/OECD/INDIGO® |

[1] Monitoring Tool for Integrated Farm Sustainability-MOTIFS [39]. [2] Indicateurs de Durabilité des Exploitations Agricoles-IDEA [15]. [3] Multicriteria Assessment of the Sustainability of Cropping Systems-MASC 2.0. [17].[4] Response Inducing Sustainability Evaluation-RISE 3.0. [64].

**Table 6.** Common ground indicators employed by INSPIA and by other sustainability assessment methods.

| | Mutual Economic Indicators with INSPIA (11) | Common Ground on Economic Dimension (%) | Mutual Social Indicators with INSPIA (4) | Common Ground on Social Dimension (%) | Mutual Environmental Indicators with INSPIA (16) | Common Ground on Environmental Dimension (%) | Common Ground of Models (%) |
|---|---|---|---|---|---|---|---|
| RISE 3.0. | 9 | 81.8 | 1 | 25.0 | 16 | 100.0 | 68.3 |
| DEXiPM | 4 | 36.4 | 1 | 25.0 | 9 | 56.3 | 38.8 |
| SAFA | 1 | 9.1 | 2 | 50.0 | 14 | 87.5 | 48.4 |
| OECD | 7 | 63.6 | 0 | 0,0 | 10 | 62.5 | 41.6 |
| SOSTARE simplified | 10 | 90.9 | 0 | 0,0 | 9 | 56.2 | 48.6 |
| Gómez-Limón and Sanchez-Fernandez | 4 | 36.4 | 1 | 25.0 | 6 | 37.5 | 32.6 |
| INDIGO® | 3 | 27.3 | 0 | 0.0 | 8 | 50.0 | 25.5 |
| MOTIFS | 5 | 45.5 | 0 | 0.0 | 8 | 50.0 | 31.5 |
| MASC | 10 | 90.9 | 0 | 0.0 | 9 | 56.3 | 48.6 |
| MASC 2.0. | 10 | 90.9 | 1 | 25.0 | 9 | 56.3 | 56.8 |
| IDEA | 8 | 72.7 | 2 | 50.0 | 10 | 62.5 | 61.1 |
| SAEMETH | 7 | 63.6 | 0 | 0.0 | 9 | 56.3 | 39.6 |
| SAFE | 9 | 81.8 | 2 | 50.0 | 7 | 43.8 | 57.9 |

According to the literature, the use of indicators and composite indices is gaining more importance in sustainability assessment and it is becoming more recognised as a tool for the adequate design of policy-making and general communication [12,20,24,26,29,46]. In this context, agricultural sustainability, in its multi-dimensional approach considers an operational development through the evaluation of the indicators system that involves the above-mentioned sustainability dimensions [73].

For building indicator-based methodologies, a current trend is to combine related indicators to obtain aggregated indicators [18,21,33,49,74,75]. However, quantifying and measuring sustainability through indicators is a complex issue [76]. This is essentially due to the heterogeneous nature of the different cropping production systems; different space-time contexts; several crop systems; possible current or future evaluations of scenarios [77].

In addition, some authors agree on the complexity of establishing indicators for sustainability assessment [78,79], and others on the interpretation of the indicators within this type of analysis [14,31,57]. This complexity is one of the main reasons why there is no agreement about the best sustainability measurement methodology [16].

The innovation and merit of INSPIA concerns the sustainability index development, the participatory weightings and aggregation procedures of the indicators, allowing respondents to convey their own preferences on weights, and it is that a multidisciplinary panel of experts (academics, technicians, researchers and farmers) have agreed not only on the basic indicators but the BMPs. The level 2 aggregated indicators are combined to define the three sustainability dimensions which compose the final composite index.

According to Table 6, and regarding the selection of the three type of indicators, the monitoring models most related to INSPIA are RISE 3.0., IDEA, SAFE, and MASC 2.0., with 68.3%; 61.1%; 57.9%; and 56.8% respectively. All of them tackle, to a greater or lesser extent, the same dimensions and scope of sustainability in INSPIA. Carpani et al., [80], illustrate that agriculture sustainability is based on an unbalanced distribution of the three principal domains (economic, social and environmental), whereas INSPIA and other approaches, for example, MOTIFS, MASC and SAEMETH, rely on the equity of the three dimensions. Therefore, this equity is inherently built into the final composite index calculation.

INSPIA indicators fully meet the requirements concerning quality criteria for their selection, initially recommended by Girardin et al., [51], and subsequently by other researchers [37,76]. INSPIA accomplishes the three steps required to meet the quality criteria on indicators validation; (i) design validation; (ii) output validation; and (iii) end-use validation, whereas other methodologies, such as INDIGO®, lack some of the quality criteria, since end-users are not questioned as part of indicator selection.

Concerning the description of indicators, the INSPIA approach, like that of SAFE, depicts a clear explanation of each of the 31 basic indicators involved [36], which confers to INSPIA a deeper soundness, given that for the majority of the rest of the reviewed tools, this description remains unknown. In agreement with other methodologies, such as MASC 2.0., most of the INSPIA basic indicators result from a simple calculation or rely upon tables. With regards to the weighting allocated to indicators, there is not much literature reflecting and explaining their weighting. Indeed, there are tools such as INDIGO® [48] which do not introduce any weighting for indicators, having just the aim of helping farmers improve their management. In this context, the INSPIA methodology presents its weightings for the indicators in a transparent way, as well as other methodologies, e.g. MOTIFS and SOSTARE.

*4.3. Assessment Methodology*

From the literature reviewed, there are already a number of indicator-based monitoring tools, that assess sustainability in agriculture [21,49,74]. Some methodologies assess sustainability at farm level like, RISE [38], IDEA, MOTIFS, SAFE and SOSTARE. In agreement with those, INSPIA also monitors sustainability at farm level. However, as stated in the literature, there is not a single measure that can accurately appraise sustainability at farm level [53,81]. Considering the evolving connotation of the word sustainability itself, it is important to show its continuous evolution, where thematic scopes, indicators and benchmarks are intended to create a constant development. Thus, sustainability assessments should be regarded as partial approaches, although they are very useful to measure and quantify sustainability in agriculture [31]. Any of the sustainability pillars (economic, social and environmental), could be broadened within INSPIA and be adapted to, or even reweighted, depending on the ongoing development of the agricultural sustainability concept.

INSPIA methodology attempts to be comprehensive and precise in the sustainability indicators accounted for in its dendrogram (Figure 5). In this regard, the moderate number of indicators composed in the INSPIA approach, and its low complexity make this method a suitable sustainability tool for technicians and farmers. Conversely, other assessment tools such as SAFA, SOSTARE and DEXiPM [32],

lead to a very complex tree of indicators. Concerning the required time to collect the data, INSPIA does not require much preparation time, as well as other reviewed approaches such as RISE [38] and IDEA. For feeding INSPIA, the required data is easily obtained from farmers' knowledge, whereas other methodologies, such as MOTIFS, require a deep knowledge [18]. In addition, the INSPIA assessment outputs are easily understood, comprehensible and decipherable by farmers, as final-users. Other approaches present a certain level of complexity to understand the final sustainability result.

### 4.3.1. Holistic Approach

Considering that the ultimate objective of the INSPIA tool is to help farmers to manage their farms sustainably, this model aims to provide a simple and forceful instrument for assessment of an individual farm. The participation of farmers as end-users in the development of the INSPIA methodology is a way to guarantee the acceptability of the method.

INSPIA, like other holistic approaches, such as MASC, RISE 3.0. or OECD, tackles many of the current agricultural challenges, such as climate change mitigation and natural capital conservation. Nevertheless, from the literature reviewed on this topic, assessment methodologies such as MOTIFS or SAFE do not pay attention to any of these relevant aspects that play a major role in the INSPIA initiative. In this context, there is a wide consensus in many of the methodologies consulted, such as DEXiPM, MASC, RISE 3.0., SAFA and OECD, that recognise the importance of measuring and assessing the impact of climate change in agriculture. Indeed, as climate change is occurring more rapidly than initially predicted [82], it is essential to track environmental indicators that monitor the dynamics of changes and trends, and to assess the reduction of emissions achievable through certain soil management practices. In fact, soil management in agriculture could be one of the best instruments to mitigate and adapt to climate change [68,83]. Certainly, farming systems, such as CA, mitigate climate change thanks to the soil organic carbon increase due to the reduction of carbon oxidation processes by reducing the intensity of tillage, and to the increase of organic matter [84–86]. Conservation agriculture also favors the adaptation of agricultural ecosystems to the negative effects of climate change by increasing crop resilience [68,87].

Being aware that there are tools like RISE, which aim to indirectly evaluate environmental impacts via management practices or through questionnaires conducted by farmers, certain threats to agricultural soil, such as those that are erosion-related, need to be addressed and more accurately assessed [38]. Despite the existence of sustainability tools like MASC 2.0. that tackle the environmental sustainability dimension, more accurate indicators in soil management are required to assess it [17]. In this sense, INSPIA addresses key soil management aspects through the basic indicators entitled 'soil tillage index', 'soil cover rate' and 'soil erosion risk'.

For its part, biodiversity is a key aspect of agricultural sustainability, and despite some authors highlighted the lack of indicators predicting the effect of some BMPs on agricultural biodiversity [40], INSPIA goes further and associates some environmental indicators which are biodiversity-related with the implementation of certain farming practices and their improvement. Both MASC approaches, as well as others such as INDIGO, do not cover biodiversity. This presents weaknesses in terms of the ability to estimate the impact of agricultural systems on biodiversity, particularly concerned with the responses of biological processes to certain farming practices and their improvement. According to Overmars et al., [3], 'soil tillage index' INSPIA indicator 16, has become an important input for monitoring the pressure of biodiversity in agriculture, since the intensity of the tillage influences agro-biodiversity.

### 4.3.2. INSPIA Thematic Scope

Concerning the thematic scope of the assessment tool, as stated by Molinos-Senante et al., [25] for the non-agricultural sector, there is no agreement on the three sustainability dimensions to reach a global approach (economic, social and environmental), to measure and evaluate sustainability among several methodologies [18] (Table 7). Most of the methodologies (DEXiPM, MOTIFS, MASC, RISE,

SAFA, SAEMETH) comprise, to a greater or lesser extent, the three dimensions, whereas others do not, such as INDIGO®, which focuses on environment and lacks both social and economic perspectives, or SOSTARE, which does not include any social indicator. There are initiatives, such as MASC, that consider the social and the biodiversity aspects that are less relevant to the cropping system scale [32], albeit others stress the need for recording those dimensions in agriculture [15]. In this regard, some coincidences are found between INSPIA and both MASC [22] and MASC 2.0. [17], that agree about considering some social sustainability aspects. INSPIA tackles a wider comprehensive approach to the family and the farmers wellbeing, through indicators that do not refer just to farm management itself, but to other family-related issues, such as farmers' satisfaction, level of training and the generational replacement. MASC 2.0. methodology also approaches those features, such as 'work overload' indicator. DEXiPM also explores and deepens the social dimension of farming practices [17,32].

There is an emerging idea in agricultural sustainability that a fourth dimension named 'institutional' or 'governance' should be envisaged [14,44,88,89]. However, in order to simplify the INSPIA methodology, this fourth dimension has not been considered.

### 4.3.3. Sector Scope

INSPIA is focused on both annual and permanent croplands. Rosnoblet et al. [90] indicate that there are many sustainability assessment methods in agriculture covering arable farming, but few of them are designed for permanent crops. In this context, there is some evidence stated by Thiollet-Scholtus and Bockstaller, [74], that INDIGO methodology, previously designed for arable systems, was amended by introducing two more environmental indicators and adapting others for the special case of vineyards.

There are different approaches concerning the scope of the models. For instance, while the INSPIA model lacks indicators for monitoring livestock farms, MOTIFS just focus on this kind of farm. There are also some models, such as IDEA or SOSTARE, which assess sustainability for mixed holdings, crops and livestock.

### 4.3.4. Geographical Scope

Most of the methodologies are not globally applicable, since they have been developed for some specific areas or countries. INSPIA's application is now limited to Belgium, Denmark, France, Germany and Spain. The reason is that the threshold of some indicators must be adapted to local conditions. Therefore, and to broaden the INSPIA scope, in countries not considered so far in the project, some basic indicators should be locally customised [53] in order to continue offering a precise reflection on farm sustainability.

### 4.3.5. INSPIA Findings

The INSPIA approach provides farmers with a sustainability pie chart, as a final visual outcome per season, and it corresponds to each aggregation step, representing the twelve existing aggregated indicators in level 1. However, as occurs in the SOSTARE model, INSPIA also offers a bar diagram representing all indicators' results for non-equivalent issues (e.g., soil organic matter, GHGs). These two ways of visualizing the results of the farms are not only very convenient and practical, communicating and presenting the overall performance, but are also useful to compare different management at farm level. Like other indicator-based monitoring tools, (e.g., MOTIFS), the advantage of these models is the ability to show an overview of farm strengths and weaknesses in a visual multilevel way, integrating the three dimensions using a pie chart, which better explains the different scenarios for each sustainability dimension of each cropping system. The more filled pie sections of the chart imply optimal values measures for aggregated indicators. INSPIA results enable farmers to know their benchmark at a given moment, and provide the basis and the guidelines to improve farm management. Likewise, another

INSPIA advantage is the feasibility of benchmarking farms thanks to the final sustainability output diagram. This feature can also be encountered in other tools, for example, MOTIFS [18].

From the literature reviewed, there are some models that are used to help assess and monitor policy performance at farm level. For instance, SOSTARE was developed by the region of Lombardia for managing the CAP until 2020. SOSTARE methodology is useful for farm advisory services to help farmers improve their economic and environmental performances, as advocated in the CAP second pillar legal proposal. Similarly, and as was indicated by Gómez-Limón and Riesgo [58], sustainability should be understood as a concept that varies in response to the needs of society. In this context, INSPIA can be modified to include advances in the shaping of the agricultural sustainability concept itself, adjusting, adding or removing the needed themes or basic indicators, or even to change benchmarks for each of the three sustainability dimensions. Given that it is possible to adjust INSPIA, it makes the tool a useful way to address new decision-making needs in future agricultural policies.

Concerning the way of giving a sustainability assessment, one of the major advantages of the INSPIA approach is that this model delivers end-users, as a targeted audience, with a composite index, whilst other models, such as MASC, INDIGO®or MOTIFs, provide no quantitative data, but simply score sustainability in different scales. INSPIA's composite index is scaled between (0–100), based on scientific rules and a robust arithmetical method (Table 7).

**Table 7.** Dashboard for different sustainability methodology frameworks. Number of basic indicators, sustainability thematic scope and type-outputs of measuring sustainability.

| Sustainability Assessment Models | Number of basic Sustainability Indicators | Sustainability Dimensions Considered Agriculture Sustainability Assessment | Agriculture Sustainability Score Measurement Type |
|---|---|---|---|
| INDIGO® | 9 (0/0/9) | Environmental dimensión | |
| OECD, 2008 | 49 (13/9/8/19) | Agriculture in the broader economic, social and environmental context/Farm management and the environment/Use of farm inputs and natural resources/Environmental impacts of agriculture | |
| SAFE | 20 (14/1/5) | Environmental/Economic/Social pillars | |
| MOTIFS | 46 (21/7/18) | Economic (33)/Social (33)/Ecological themes (33) | 0 (Non sustainable) 100 (sustainable) per theme |
| IDEA | 41 (19/16/6) | Agro-ecological/Socio-territorial/Economic scale | Score between (0–100) |
| MASC | 32 (4/5/23) | Economic (33)/Social (33)/Environmental sustainability (33) | Very low/Low/Medium/High/Very high |
| MASC 2.0. | 39 (12/7/20) | Economic (33)/Social (33)/Environmental sustainability (33) | Very low/Low/Medium/High/Very high |
| Gómez-Limón and Sanchez-Fernandez, 2010 | 16 (3/4/9) | Economic/Social/Environmental function | |
| DEXiPM | 45 (6/18/21) | Economic/Social/Environmental sustainability | Very high/High/Medium/Low/Very low |
| RISE | 12 (7/4/1) | Ecological/Economical/Social | Value (−100–+100) |
| RISE 3.0. | | Soil use/Animal husbandry/Material use and environmental protection/Water use/Energy and Climate/Biodiversity/Working conditions/Quality of life/Economic viability/Farm management % unknown | Problematic (0–33)/Critical (34–66)/Positive (67–100) |
| SAFA | 116 (19/52/26/19) | Governance/Environmental/Economic/Social dimensions | Best (>80%)/Good (60–80)%/Moderate (40–60)%/Limited (20–40)%/Unacceptable (<20%) |
| SAEMETH | | Socio-cultural (33%)/Agro-environmental (33%)/Economic dimensions (33%) | |
| SOSTARE | 125 (92/27/6) | Agronomy/Economy/Ecology | Separately indicators referenced to thresholds |

## 5. Conclusions

This article presents the INSPIA methodology for assessing sustainability in agriculture. INSPIA is based on the implementation of 15 best management practices, contributing to 31 indicators. The INSPIA model is applicable to both annual and permanent crops.

The in-field application of INSPIA's best management practices has been validated on 59 farms in several European countries. This methodology helps farmers improve their economic, social and environmental performance, as advocated in strategic agro-environmental policies such as the European Common Agricultural Policy.

INSPIA has been discussed and compared to other initiatives, and the result of this process shows it to be a robust methodology that can be adapted to different agro-climatic regions. For monitoring the impacts of different farming practices and systems, INSPIA meets the key standards and is therefore a useful and valid tool to support decision-making by agricultural, environmental and social-welfare policy makers.

**Author Contributions:** The authors' contributions are as follows: conceptualisation, E.J.G.-S., P.T.-T. and M.R.G.-A.; methodology, P.T.-T. and M.R.G.-A, original draft preparation and writing, P.T.-T and E.J.G.-S.; review and editing, P.T.-T., E.J.G.-S. and G.B.; supervision and validation, P.T.-T., M.R.G.-A., G.B. and E.J.G.-S.

**Funding:** This research was supported by the European Crop Protection Association (ECPA) and the European Agriculture Conservation Federation (ECAF).

**Acknowledgments:** The authors want to thank the following organisations: Asociación Española Agricultura de Conservación Suelos Vivos (AEACSV) and Association pour la Promotion d'une Agriculture Durable (APAD) for putting the ideas into practice on farms; support from Gesellschaft für konservierende Bodenbearbeitung e.V. (GKB) and the Belgian Institute for Agricultural and Fisheries Research (ILVO); and to the European Conservation Agriculture Federation (ECAF) and the European Crop Protection Association (ECPA) for co-sponsoring the INSPIA project. We would also like to thank the individuals who are too numerous to name for their help: the experts consulted on the development of the INSPIA methodology, and the steering team for helping guide the project.

**Conflicts of Interest:** The authors declare no conflict of interest.

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
