# Peer review of "Sustainability Assessment of Annual and Permanent Crops: The Inspia Model"

_sustainability, doi:10.3390/su11030738_

Round 1
Reviewer 1 Report
The objectives of the project, justification and description of processes, presentation of results, and comparisons with alternative approaches are all clearly explained and defended in the manuscript. However, the project suffer from the same deficits as other similar projects--including those chosen for comparison. First, the sustainability or lack thereof, of farming any endeavor cannot be reduced to a single quantifiable indicator. Such indicators, at best, can only indicate farms that are following a select set of indicators "related to sustainability" better than are others. Second, the social indicators in this project, while at least considered, are very weak--indicating only farm level considerations. Socially sustainable farms must produce wholesome nutritious food, protect public health from agricultural relate risks, contribute to food security my means other than simple productivity, and contribute to quality of life in rural communities--in addition to providing quality employment opportunities for farmers. Finally, the linkage between the indicators in this project and the UN Sustainable Development Goal are very tenuous at best. That being said, the manuscript does appear to make a significant positive contribution to a substantial body of literature that is subject to the same limitations.
Author Response
First of all, we would like to thank you for going through the manuscript, and make some suggestions and comments. We will take all of them into account for future research.
Reviewer 2 Report
The paper is well developed and addresses the issue of sustainability in agriculture, one of the topics currently most studied by the scientific community of the sector.
The approach is very interesting and can be a useful decision support tool for farmers and technicians.
I suggest that the authors further investigate the state of the art of integrated sustainability assessment in agro-industry, particularly the literature on Life Cycle Sustainability Assessment (LCSA), where research is making great advances.
Only as example:
De Luca, A.I., Falcone, G., Stillitano, T., Iofrida, N., Strano, A., Gulisano, G. (2018) Evaluation of sustainable innovations in olive growing systems: A Life Cycle Sustainability Assessment case study in southern Italy. Journal of Cleaner Production, 171, pp. 1187-1202. DOI: 10.1016/j.jclepro.2017.10.119
Notarnicola B., Sala S., Anton A., McLaren S.J., Saouter E., Sonesson U., (2017) The role of life cycle assessment in supporting sustainable agri-food systems: A review of the challenges. Journal Cleaner Production, 140 (2), 399–409. DOI: 10.1016/j.jclepro.2016.06.071
De Luca, A.I., Iofrida, N., Leskinen, P., Stillitano, T., Falcone, G., Strano, A., Gulisano, G. (2017) Life cycle tools combined with multi-criteria and participatory methods for agricultural sustainability: Insights from a systematic and critical review). Science of the Total Environment, 595, pp. 352-370. DOI: 10.1016/j.scitotenv.2017.03.284
Ren, J., Manzardo, A., Mazzi, A., Zuliani, F., Scipioni, A. (2015) Prioritization of bioethanol production pathways in China based on life cycle sustainability assessment and multicriteria decision-making. International Journal of Life Cycle Assessment, 20 (6), pp. 842-853. DOI: 10.1007/s11367-015-0877-8
Zamagni A., (2012). Life cycle sustainability assessment. International Journal of Life Cycle Assessessment. 17(4). DOI: 10.1007/s11367-012-0389-8
Author Response
First of all, I would like to thank you for going through the manuscript, and make the suggestions and comments concerning the Life Cycle Assessment tools, that indeed, research is making a great progress.
In this regards, some interesting references have been added to the manuscript in the introduction section.
